# Serum Osteopontin Level Is Positively Associated with Aortic Stiffness in Patients with Peritoneal Dialysis

**DOI:** 10.3390/life12030397

**Published:** 2022-03-09

**Authors:** Kai-Hsiang Chang, Chih-Hsien Wang, Bang-Gee Hsu, Jen-Pi Tsai

**Affiliations:** 1Division of Nephrology, Hualien Tzu Chi Hospital, Buddhist Tzu Chi Medical Foundation, Hualien 97004, Taiwan; 102311106@gms.tcu.edu.tw (K.-H.C.); wangch33@tzuchi.com.tw (C.-H.W.); 2School of Medicine, Tzu Chi University, Hualien 97004, Taiwan; 3Division of Nephrology, Department of Internal Medicine, Dalin Tzu Chi Hospital, Buddhist Tzu Chi Medical Foundation, Chiayi 62247, Taiwan

**Keywords:** arterial stiffness, carotid–femoral pulse wave velocity, osteopontin, peritoneal dialysis

## Abstract

Background: Osteopontin (OPN) is regarded as a proinflammatory and proatherogenic molecule related to atherosclerosis. We aimed to evaluate the relationship between serum OPN and aortic stiffness (AS) of peritoneal dialysis (PD) patients. Methods: OPN and carotid-femoral pulse wave velocity (cfPWV) were measured by a commercial enzyme-linked immunosorbent assay kit and a validated tonometry system, respectively. Patients with cfPWV > 10 m/s were designated into the AS group. Results: Twenty-two patients (31.4%) were segregated into the AS group. Multivariate linear and logistic regression analysis showed that OPN was significantly related to cfPWV and was an independent predictor of AS. The receiver operating characteristic curve analysis showed that OPN was correlated with AS with an area under the curve of 0.903 (95% CI 0.809–0.961, *p* < 0.001). Conclusions: For PD patients, the serum OPN level was correlated with cfPWV and could play an important role in the process of AS.

## 1. Introduction

Cardiovascular diseases (CVD), which may be caused by traditional risk factors, such as diabetes mellitus (DM) and hypertension (HTN), as well as other specific risk factors, have been reported to be related with the worse prognosis of patients with chronic kidney disease (CKD) [1]. Aortic stiffness (AS), which is caused by a dysregulated expression of elastin and collagen, oxidative stress, low grade inflammation, and abnormal mineral metabolism, is one of the multiple characteristics of CVD, among the other alterations, such as coronary artery calcification, vessel wall thickening, fatty streak formation, and atherosclerotic plaques. AS can result in decreased coronary artery perfusion pressure, deterioration of renal function, and future CVD in patients with CKD [2,3,4,5,6], and could represent subclinical organ damage. The carotid–femoral pulse wave velocity (cfPWV) is a well-known noninvasive technique that indicates vascular function, and is a strong predictor of cardiovascular events and mortality, as well as all-cause mortality in patients with CKD and end stage renal disease (ESRD), independent of the traditional CV risk factors [5,7]. Evidence has shown that patients with CKD with high cfPWV are more likely to have worse renal function, ESRD, or death [6].

Osteopontin (OPN) was initially found to be a calcium-binding glycophosphoprotein that is involved in bone remodeling, and was later known to have multiple roles in promoting the inflammatory process, such as cell recruitment to inflammatory sites, cell attachment, and wound healing. It was reported to be a mediator in regulating the pathogenesis of sepsis, and had a role in predicting the survival or recovery from renal replacement therapy in patients with acute kidney injury [8,9,10]. In addition, OPN was found to have an inverse association with renal function, and after being identified in the arterial tissue to be related with atherosclerotic plaques, OPN was further known to be involved in arterial smooth muscle tissue phenotypic transition as a modulator of atherosclerosis and arterial calcification [11,12,13,14,15]. Furthermore, evidence has shown that serum OPN levels are correlated with central AS in a geriatric population, as well as in patients with CKD and coronary artery disease (CAD) [11,12,16,17,18].

Although OPN has been considered to have a role in a variety of diseases, its relationship with AS in patients on hemodialysis (HD) had been controversial [19]. Moreover, there are no studies on the association of OPN levels and AS in patients on peritoneal dialysis (PD). We aimed to study the role of serum OPN in the development of AS by measuring cfPWV in patients on PD.

## 2. Materials and Methods

### 2.1. Patients

From June 2015 to October 2016, with approval from the Protection of Human Subjects Institutional Review Board (IRB103-136-B), patients who underwent regular PD for more than three months at a medical center were included. Participants were excluded if they refused to sign informed consent or had an acute illness, such as infection, malignancy, stroke, heart failure, acute coronary syndrome, or amputation, at the time of blood sampling. All participants provided informed consent before participating in this study.

Data on the adequacy of dialysis, which included the weekly fractional clearance index for urea (Kt/V), total clearance of creatinine, and peritoneal clearance of creatinine, were retrospectively reviewed and collected from the medical records. The morning systolic blood pressure (SBP) and diastolic BP (DBP) of all patients were measured three times at five-minute intervals by trained staff using standard mercury sphygmomanometers with appropriate cuff sizes and after having the patient rest for at least 10 min. HTN was defined as SBP ≥ 140 mmHg and/or DBP ≥ 90 mmHg, or as a receipt of any antihypertensive medication in the past two weeks. DM was defined as fasting plasma glucose > 126 mg/dL or the use of oral hypoglycemic medications or insulin.

A total of 70 patients on PD were enrolled in this study. Of these patients, 50 received continuous ambulatory PD (CAPD, Dianeal, Baxter Health Care, Taiwan), with three to five dialysate exchanges per day; the 20 remaining patients underwent four to five dialysate exchanges each night with an automated device (i.e., automated PD (APD)).

### 2.2. Anthropometric Analysis

Body mass index (BMI) was calculated as the body weight in kilograms divided by the square of the height in meters. It was measured individually to the nearest half centimeter and to the nearest half kilogram, with each patient wearing light clothing and no shoes.

### 2.3. Biochemical Investigations

In the morning before exchanging the dialysate, approximately 5 mL of blood sample was collected from each participant and was immediately centrifuged at 3000× *g* for 10 min. Within 1 h of collection, the serum samples were stored at 4 °C and sent for biochemical analyses. Serum levels of blood urea nitrogen, creatinine, fasting glucose, albumin, total cholesterol, triglyceride, total calcium, phosphorus, and C-reactive protein (CRP) were measured using an autoanalyzer (Siemens Advia 1800, Siemens Healthcare GmbH, Henkestr, Germany). The serum levels of the OPN (catalog number: BMS2066; eBioscience Inc., San Diego, CA, USA) and intact parathyroid hormone (iPTH) (catalog number: NM59041; IBL International, Hamburg, Germany) were quantified by commercially available enzyme-linked immunosorbent assays [20].

### 2.4. Carotid–Femoral Pulse Wave Velocity Measurements

The cfPWV was transcutaneously measured by recording the arterial pressure pulse waveform on applanation tonometry (SphygmoCor system, AtCor Medical, New South Wales, Australia), as previously described [20]. Measurements were performed in the morning, with each participant in the supine position and after resting for at least 10 min in a quiet and temperature-controlled room. Records were made simultaneously with an electrocardiogram signal, which provided a reference for the R wave. Pulse wave recordings were performed consecutively on two superficial artery sites (i.e., carotid–femoral segment). Integral software was used to process each set of pulse wave recordings and electrocardiogram data, and to calculate the mean time difference between the R wave and the pulse wave on a beat-to-beat basis, with an average of 10 consecutive cardiac cycles. The cfPWV was calculated using the distance and mean time difference between the two recorded points. Quality indices, which were included in the software, were set to ensure uniformity of data. The study population was divided into the high AS and control groups, based on the cfPWV values of >10 m/s and ≤10 m/s, respectively, according to the European Society of Hypertension and of the European Society of Cardiology guidelines [21].

### 2.5. Statistical Analysis

Continuous variables were analyzed for normal distribution using the Kolmogorov–Smirnov test. Variables that had a non-normal distribution were logarithmically transformed for linear regression analysis. Normally distributed continuous variables were expressed as mean ± standard deviation (SD) and were compared between the groups using two-tailed independent Student’s *t*-test. Variables with non-normal distribution (i.e., fasting glucose, iPTH, CRP, and OPN) were expressed as median and interquartile range (IQR), and were compared between the groups using the Mann–Whitney U-test. Categorical variables were expressed as number (percentage) and were analyzed by the χ^2^ test. The risk factors for developing AS were determined by multivariate logistic regression analysis. The associations of cfPWV with the variables were analyzed by simple and multivariate forward stepwise linear regression analysis. The receiver operating curve (ROC) was used to calculate the area under the curve (AUC) to identify the optimal cutoff value of OPN so as to predict a high AS in patients on PD. Data were analyzed using SPSS for Windows (version 19.0; SPSS Inc., Chicago, IL, USA). A *p* value of <0.05 was considered statistically significant.

## 3. Results

The clinical characteristics of the 70 patients included in this study are presented in Table 1; 40 (57.1%) were women and 50 (71.4%) were on CAPD. There were 31 (44.3%) and 58 (82.9%) patients who had DM and HTN, respectively. There were 20 patients (29.4%) included in the high AS group. Compared with the patients in the control group, those in the high AS group were older (*p* = 0.001), had longer PD vintage (*p* = 0.024), and had higher serum levels of CRP (*p* = 0.005) and OPN (*p* < 0.001), but had a similar proportion of men and women, PD model, BMI, frequency of coexisting DM and HTN, adequacy of dialysis, and medications used, including the usage of calcium carbonate, calcitriol, and Icodextrin.

For the multivariate logistic regression analysis, the independent risk factors for the development of AS in patients on PD were serum OPN (OR 1.044, 95% CI 1.020–1.069 (*p* < 0.001) and PD vintage (OR 1.027, 95% CI 1.007–1.047, *p* = 0.009) (Table 2). For the ROC curve analysis, the optimal value of serum OPN to predict AS was 39.67 ng/mL, with a sensitivity of 86.36% (95% C.I. 65.1–97.1%), specificity of 91.67% (95% C.I. 80.0–97.7%), and AUC of 0.903 (95% CI 0.809–0.961, *p* < 0.001) (Figure 1).

For the simple linear regression analysis, cfPWV was positively correlated with age (*r* = 0.394, *p* = 0.001), logarithmically transformed (log) CRP (*r* = 0.375, *p* = 0.001), and serum log-OPN level (*r* = 0.529, *p* < 0.001). After adjusting for these covariates by multivariable forward stepwise linear regression analysis, the log-CRP (β = 0.290, adjusted R^2^ change = 0.073, *p* = 0.005) and log-OPN levels (β = 0.477, adjusted R^2^ change = 0.269, *p* < 0.001) were found to be independently correlated with cfPWV (Table 3).

## 4. Discussion

This study showed that patients on PD and with high OPN levels had a high risk of developing AS. In addition, OPN levels could be a predictor of AS, independent of age, sex, comorbidities, and medications used.

Aside from the traditional CV risk factors, including DM, hyperlipidemia, and elevated BMI, as well as mineral metabolism derangements that cause elastin fragmentation and medial layer calcification in CKD, aging might contribute to the structural and functional changes in vessels and have been shown to be related with the development of AS [5,22,23]. In a study conducted on patients on HD, age-associated increase in aortic PWV, increase in aortic bifurcation diameter, decrease in aortic tapering, and relatively low brachial/AS gradient that declined more significantly with age were found [23]. Inflammation has been known to play a role in the process of vascular calcification, based on reports that high serum CRP, old age, and DM are associated with the presence of aortic calcification in patients with CKD and on HD [24,25]. In addition, the glucose degradation and advanced glycation end products from the PD solution were considered to be associated with microvascular sclerosis, based on evidence that showed a positive association between cfPWV and a relatively long duration on PD [26,27,28]. In this study, we found that advanced age and a relatively high CRP level in patients on PD were associated with a relatively high cfPWV. Furthermore, after adjusting for confounders, receipt of PD for a relatively long period was an independent predictor for the development of AS.

OPN is a hyperphosphorylated extracellular matrix protein that has a variety of functions in the immune system, cell viability or proliferation, and wound healing. It is a key factor in atherosclerosis and is now being regarded as an important mediator of atherosclerotic plaque formation and arterial calcification [14,15,29,30]. In normal kidneys, OPN is mainly found in the loop of Henle and distal renal tubules, and is upregulated in all tubules and glomeruli after renal damage, such as nephrolithiasis, acute kidney injury, and glomerulonephritis [31]. Studies showed that serum levels of OPN increased in parallel with the decline of renal function and were positively correlated with SBP, the homeostasis model assessment of insulin resistance, and the presence of dyslipidemia or carotid artery intima–media thickening [11,12,13,32,33]. In a longitudinal study on 3500 patients with CAD, plasma OPN was shown to significantly predict future CV death, nonfatal myocardial infarction, and hospitalization for heart failure [34]. Moreover, in patients with CVD or peripheral artery disease, the level of OPN was reported to be a predictor of long-term adverse outcomes, independent of other risk factors [35]. An in vivo study showed accelerated and enhanced arterial calcification in matrix Gla protein (MGP) and OPN double-deficient mice (MGP−/−OPN−/−) than in mice deficient in MGP alone (MGP−/− OPN+/+); this result implied the role of OPN in mediating the process of arterial calcification [36]. In terms of vascular function, the serum OPN level was positively associated with cfPWV, along with impaired flow-mediated dilation, disease severity, and increased interleukin 6 in patients with CAD [16,18]. In a cross-sectional study that enrolled 1003 patients with DM, the OPN level was associated with a relatively low ankle–brachial index and relatively high arterial stiffness summary score; this indicated the role of OPN in the process of central and peripheral arterial stiffness [37]. In the present study, we found that patients on PD with a high serum OPN level had high risk of developing AS. However, a study on a small number of patients on HD failed to demonstrate the association of AS or cfPWV with OPN [19]. Taken together, the underlying role of high circulating serum levels of OPN in patients with CKD and ESRD with AS remains to be elucidated. Nevertheless, OPN might represent a bridge between inflammation and the resultant vascular damage, or may counter-regulate the uremic calcification process [19,29,38].

The limitation of this study was its cross-sectional design with a limited number of patients on PD. Therefore, the causal relationship of serum OPN with AS and the role of OPN in the development of AS in patients on PD should be confirmed by further longitudinal studies.

## 5. Conclusions

In conclusion, along with old age, high CRP, and long duration on PD, the serum OPN level could be a novel biomarker for predicting the development of AS in patients on PD. OPN might modulate the process of AS, but the mechanism remains to be studied.

## Figures and Tables

**Figure 1 life-12-00397-f001:**
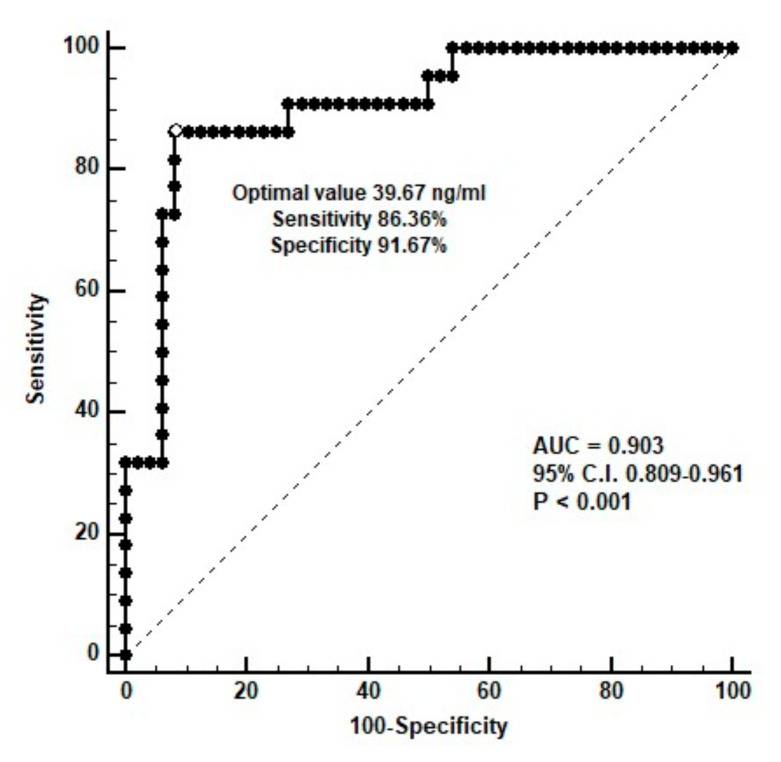
The area under the receiver operating characteristic curve indicates the diagnostic power of the serum osteopontin level for predicting aortic stiffness among 70 peritoneal dialysis patients.

**Table 1 life-12-00397-t001:** Clinical variables of the 70 peritoneal dialysis patients with or without aortic stiffness.

Characteristic	All Participants (*n* = 70)	Control Group (*n* = 48)	Aortic Stiffness Group (*n* = 22)	*p* Value
Age (years)	56.86 ± 15.11	52.96 ± 15.20	65.36 ± 11.10	0.001 *
Peritoneal dialysis vintage (months)	50.26 ± 40.87	42.88 ± 38.93	66.36 ± 41.22	0.024 *
Height (cm)	159.89 ± 8.58	160.71 ± 8.60	158.09 ± 8.47	0.239
Body weight (kg)	63.35 ± 13.64	63.50 ± 12.55	63.03 ± 16.07	0.895
Body mass index (kg/m^2^)	24.66 ± 4.14	24.57 ± 4.08	24.86 ± 4.41	0.785
Carotid-femoral PWV (m/s)	9.08 ± 3.15	7.38 ± 1.84	12.80 ± 1.95	<0.001 *
Systolic blood pressure (mmHg)	142.33 ± 23.69	140.31 ± 25.13	146.73 ± 20.01	0.296
Diastolic blood pressure (mmHg)	84.64 ± 12.94	82.96 ± 13.45	88.32 ± 11.16	0.108
Total cholesterol (mg/dL)	167.53 ± 38.47	164.35 ± 36.51	174.45 ± 42.49	0.311
Triglyceride (mg/dL)	178.66 ± 115.53	184.13 ± 129.86	166.73 ± 76.77	0.562
Fasting glucose (mg/dL)	106.00 (95.00–131.50)	103.50 (93.25–124.25)	114.00 (96.75–149.00)	0.086
Albumin (g/dL)	3.73 ± 0.35	3.74 ± 0.38	3.70 ± 0.28	0.662
Blood urea nitrogen (mg/dL)	57.77 ± 18.27	58.17 ± 18.92	56.91 ± 17.16	0.791
Creatinine (mg/dL)	11.00 ± 3.01	11.07 ± 3.12	10.83 ± 2.85	0.756
Total calcium (mg/dL)	9.11 ± 0.79	9.08 ± 0.73	9.18 ± 0.90	0.617
Phosphorus (mg/dL)	5.21 ± 1.40	5.26 ± 1.42	5.10 ± 1.38	0.673
Intact parathyroid hormone (pg/mL)	248.63 (113.37–522.48)	257.35 (121.64–546.23)	221.77 (87.36–549.23)	0.899
C reactive protein (mg/dL)	0.26 (0.07–0.73)	0.14 (0.06–0.52)	0.32 (0.25–1.18)	0.005 *
Osteopontin (ng/mL)	20.36 (9.85–56.95)	13.36 (7.43–24.36)	74.53 (41.77–141.66)	<0.001 *
Weekly Kt/V	2.10 ± 0.39	2.15 ± 0.45	1.95 ± 0.33	0.070
Peritoneal Kt/V	1.74 ± 0.48	1.76 ± 0.46	1.71 ± 0.42	0.654
Total clearance of creatinine (L/week)	59.89 ± 24.85	61.54 ± 25.66	55.09 ± 20.20	0.308
Peritoneal clearance of creatinine (L/week)	41.15 ± 16.16	42.33 ± 16.88	42.35 ± 15.35	0.995
Female, *n* (%)	40 (57.1)	28 (58.3)	12 (54.5)	0.766
Diabetes, *n* (%)	31 (44.3)	22 (45.8)	9 (40.9)	0.700
Hypertension, *n* (%)	58 (82.9)	39 (81.3)	19 (86.4)	0.598
CAPD, *n* (%)	50 (71.4)	33 (68.8)	17 (77.3)	0.464
ACE inhibitor use, *n* (%)	5 (7.1)	4 (8.3)	1 (4.5)	0.568
ARB use, *n* (%)	28 (40.0)	19 (39.6)	9 (40.9)	0.916
β-blocker use, *n* (%)	27 (38.6)	20 (41.7)	7 (31.8)	0.432
CCB use, *n* (%)	30 (42.9)	20 (41.7)	10 (45.5)	0.766
Statin use, *n* (%)	22 (31.4)	16 (33.3)	6 (27.3)	0.612
Fibrate use, *n* (%)	2 (2.9)	1 (2.1)	1 (4.5)	0.566
Calcium carbonate use, *n* (%)	37 (52.9)	27 (56.3)	10 (45.5)	0.401
Calcitriol use, *n* (%)	20 (28.6)	14 (29.2)	6 (27.3)	0.871
Icodextrin, *n* (%)	48 (68.6)	32 (66.7)	16 (72.7)	0.612

Values for continuous variables are shown as mean ± standard deviation after analysis by Student’s *t*-test; variables not normally distributed are shown as median and interquartile range after analysis by the Mann–Whitney U test; values are presented as number (%) and analysis after analysis by the Chi-squared test. CAPD, continuous ambulatory peritoneal dialysis; weekly Kt/V, weekly fractional clearance index for urea; ACE, angiotensin-converting enzyme; ARB, angiotensin-receptor blocker; CCB, calcium-channel blocker. * *p* < 0.05 was considered statistically significant.

**Table 2 life-12-00397-t002:** Multivariate logistic regression analysis of the factors correlated to aortic stiffness among 70 peritoneal dialysis patients.

Variables	Odds Ratio	95% Confidence Interval	*p* Value
Osteopontin, 1 ng/mL	1.044	1.020–1.069	<0.001 *
Peritoneal dialysis vintage, 1 month	1.027	1.007–1.047	0.009 *
Age, 1 year	1.063	0.983–1.150	0.127
C reactive protein, 1 mg/dL	1.762	0.720–4.313	0.215

Analysis of the data was done using the multivariate logistic regression analysis (adopted factors: age, peritoneal dialysis vintage, C reactive protein, and osteopontin). * *p* < 0.05 was considered statistically significant.

**Table 3 life-12-00397-t003:** Correlation between central pulse wave velocity and clinical variables among 70 peritoneal dialysis patients.

Variables	Central PWV (m/s)				
	Simple Linear Regression		Multivariate Linear Regression		
	*r*	*p* Value	Beta	Adjusted R^2^ Change	*p* Value
Female	−0.071	0.557	–	–	–
Diabetes mellitus	0.016	0.896	–	–	–
Hypertension	−0.083	0.497	–	–	–
Age (years)	0.394	0.001 *	–	–	–
Peritoneal dialysis vintage (months)	0.221	0.066	–	–	–
Height (cm)	−0.034	0.782	–	–	–
Body weight (kg)	0.038	0.754	–	–	–
Body mass index (kg/m^2^)	0.010	0.932	–	–	–
Systolic blood pressure (mmHg)	0.038	0.752	–	–	–
Diastolic blood pressure (mmHg)	0.037	0.762	–	–	–
Total cholesterol (mg/dL)	0.105	0.387	–	–	–
Triglyceride (mg/dL)	−0.002	0.984	–	–	–
Log-Glucose (mg/dL)	0.095	0.435	–	–	–
Albumin (g/dL)	−0.069	0.573	–	–	–
Blood urea nitrogen (mg/dL)	0.047	0.697	–	–	–
Creatinine (mg/dL)	−0.099	0.415	–	–	–
Total calcium (mg/dL)	0.051	0.674	–	–	–
Phosphorus (mg/dL)	−0.105	0.385	–	–	–
Log-iPTH (pg/mL)	0.004	0.971	–	–	–
Log-CRP (mg/dL)	0.375	0.001 *	0.290	0.073	0.005 *
Log-Osteopontin (ng/mL)	0.529	<0.001 *	0.477	0.269	<0.001 *
Weekly Kt/V	−0.168	0.164	–	–	–
Peritoneal Kt/V	−0.199	0.098	–	–	–
Total clearance of creatinine (L/week)	0.056	0.648	–	–	–
Peritoneal clearance of creatinine (L/week)	−0.073	0.547	–	–	–

Data of glucose, iPTH, CRP, and osteopontin levels showed skewed distribution, and therefore were log-transformed before the analysis. * *p* < 0.05 was considered statistically significant. Analysis of the data was done using the univariate linear regression analyses or multivariate stepwise linear regression analysis (adopted factors: age, log-CRP, and log-osteopontin). iPTH, intact parathyroid hormone; CRP, C reactive protein; weekly Kt/V, weekly fractional clearance index for urea.

## Data Availability

The data presented in this study are available upon request from the corresponding author.

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
