# Peer review of "Serum Osteopontin Level Is Positively Associated with Aortic Stiffness in Patients with Peritoneal Dialysis"

_life, 2022, doi:10.3390/life12030397_

Round 1

Reviewer 1 Report

Comments to the author:

Chang et al. reported about the relationship between the serum osteopontin level and the aortic stiffness in peritoneal dialysis (PD) patients. 
Although the manuscript is well written and includes some important new findings, there are some issues to be addressed.

1. The information about phosphate binder (in particulare Calcium carbonate) or vitamin D use should be described.

2. Moreover, the information about icodexitrin dialysate use also should be described.

Author Response

Comments to the author:

Chang et al. reported about the relationship between the serum osteopontin level and the aortic stiffness in peritoneal dialysis (PD) patients. 
Although the manuscript is well written and includes some important new findings, there are some issues to be addressed.

  1. The information about phosphate binder (in particulare Calcium carbonate) or vitamin D use should be described.

Ans: Thanks for your comments. We added the information about phosphate binder and usage of vitamin D in Table 1.

  1. Moreover, the information about icodexitrin dialysate use also should be described.

Ans: Thanks for your comments. We add the information about the usage of icodextran in Table 1.

Reviewer 2 Report

The manuscript by Chang et al, entitled “Serum osteopontin level is positively associated with aortic stiffness in patients with peritoneal dialysis” has investigated relationship between serum OPN and aortic stiffness of peritoneal dialysis patients. This paper found that patients on peritoneal dialysis and had high OPN levels had a high risk of developing aortic stiffness. They concluded that serum OPN level could be a predictor of aortic stiffness. This paper is well-written, interesting manuscript. I have the following minor comments.

Comments

  1. Figure 1. Please describe the AUC value in the figure or in the text.
  2. Figure 1. Please remove the word “Osteopontin (ng/ml)” from the figure to avoid reader’s confusing.

Author Response

Comments and Suggestions for Authors

The manuscript by Chang et al, entitled “Serum osteopontin level is positively associated with aortic stiffness in patients with peritoneal dialysis” has investigated relationship between serum OPN and aortic stiffness of peritoneal dialysis patients. This paper found that patients on peritoneal dialysis and had high OPN levels had a high risk of developing aortic stiffness. They concluded that serum OPN level could be a predictor of aortic stiffness. This paper is well-written, interesting manuscript. I have the following minor comments.

Comments

  1. Figure 1. Please describe the AUC value in the figure or in the text.

Ans: Thanks for comments. We revised the description of optimal value of osteopontin with value of AUC of the manuscript as “Of ROC curve analysis, the optimal value of erum OPN to predict AS was 39.67 ng/ml with sensitivity of 86.36% (95% C.I. 65.1%-97.1%), specificity of 91.67% (95% C.I. 80.0%-97.7%) and AUC of 0.903 (95% CI 0.809–0.961, p < 0.001)” and Figure 1 as below.

  1. Figure 1. Please remove the word “Osteopontin (ng/ml)” from the figure to avoid reader’s confusing.

Ans: Thanks for your comments. We revised the Figure 1 according to your recommendation.

Round 2

Reviewer 1 Report

N/A